# Transcriptome Analyses Identify Potential Key microRNAs and Their Target Genes Contributing to Ovarian Reserve

**DOI:** 10.3390/ijms221910819

**Published:** 2021-10-06

**Authors:** Yoon-Young Kim, Kwang-Soo Kim, Yong-Jin Kim, Sung-Woo Kim, Hoon Kim, Seung-Yup Ku

**Affiliations:** 1Department of Obstetrics and Gynecology, Seoul National University Hospital, Seoul 03080, Korea; yoonykim@snu.ac.kr (Y.-Y.K.); byulbi@naver.com (S.-W.K.); obgyhoon@gmail.com (H.K.); 2Institute of Reproductive Medicine and Population, Medical Research Center, Seoul National University, Seoul 03080, Korea; 3Transdisciplinary Department of Medicine & Advanced Technology, Seoul National University Hospital, Seoul 03080, Korea; kksoo716@gmail.com; 4Department of Obstetrics and Gynecology, Korea University College of Medicine, Goryeodae-ro 73, Seongbuk-gu, Seoul 02841, Korea; zinigo@korea.ac.kr

**Keywords:** microRNA, RNA-sequencing, differentially expressed genes, *Piwil*

## Abstract

Female endocrinological symptoms, such as premature ovarian inefficiency (POI) are caused by diminished ovarian reserve and chemotherapy. The etiology of POI remains unknown, but this can lead to infertility. This has accelerated the search for master regulator genes or other molecules that contribute as enhancers or silencers. The impact of regulatory microRNAs (miRNAs) on POI has gained attention; however, their regulatory function in this condition is not well known. RNA sequencing was performed at four stages, 2-(2 W), 6-(6 W), 15-(15 W), and 20-(20 W) weeks, on ovarian tissue samples and 5058 differentially expressed genes (DEGs) were identified. Gene expression and enrichment were analyzed based on the gene ontology and KEGG databases, and their association with other proteins was assessed using the STRING database. Gene set enrichment analysis was performed to identify the key target genes. The DEGs were most highly enriched in 6 W and 15 W groups. *Figla*, *GDF9*, *Nobox*, and *Pou51* were significantly in-creased at 2 W compared with levels at 6 W and 20 W, whereas the expression of *Foxo1*, *Inha*, and *Taf4b* was significantly de-creased at 20 W. *Ccnd2* and *Igf1* expression was maintained at similar levels in each stage. In total, 27 genes were upregulated and 26 genes interacted with miRNAs; moreover, stage-specific upregulated and downregulated interactions were demonstrated. Increased and decreased miRNAs were identified at each stage in the ovaries. The constitutively expressed genes, *Ccnd2* and *Igf1*, were identified as the major targets of many miRNAs (*p* < 0.05), and *Fshr* and *Foxo3* interacted with miRNAs, namely mmu-miR-670-3p and mmu-miR-153-3p. miR-26a-5p interacted with *Piwil2*, and its target genes were downregulated in the 20 W mouse ovary. In this study, we aimed to identify key miRNAs and their target genes encompassing the reproductive span of mouse ovaries using mRNA and miRNA sequencing. These results indicated that gene sets are regulated in the reproductive stage-specific manner via interaction with miRNAs. Furthermore, consistent expression of *Ccnd2* and *Igf1* is considered crucial for the ovarian reserve and is regulated by many interactive miRNAs.

## 1. Introduction

Diminished ovarian reserve, such as premature ovarian inefficiency (POI) is an increasing problem and leads to infertility in women under the age of 40 years. In previous studies, genes involved in POI have been identified (Table 1), but the critical regulators of this condition have not been fully elucidated. In addition to regulatory genes, non-coding RNAs are considered candidate regulators. miRNAs are well-known conserved regulators of gene expression [1], and their contribution to germ cell developmental control has been studied. Their involvement in ovarian follicles and oocyte development has also been demonstrated [2,3,4]. The correlation between miRNAs and POI has been studied, and several miRNAs have been suggested to be associated with POI [5]. Among these, miR-23a and miR-27a are involved in granulosa cell proliferation and apoptosis by controlling *XIAP* [6,7] and *IGFBP2* [8].

The fundamental reasons for POI are not fully elucidated due to its complex regulation; however, advancements in array technology have enhanced the identification of major genes and other regulators, such as non-coding RNAs, including microRNAs (miRNAs) [9,10] and long non-coding RNAs [11,12]. miRNAs are small non-coding RNAs that contribute to the regulation of gene expression at transcriptional and post-transcriptional levels [1,13]. Their regulatory functions in the ovary are also known [14,15], and their regulatory role [3,16] and expression in patients with POI have been studied [17,18].

The reproductive span of oocytes changes dramatically during the life cycle of women [19]. The number of oocytes in a female is pre-determined, and these are not reproducible and are diminished until menopause via menstrual cycles [20]. However, the reproductive cycle becomes irregular with endocrinological symptoms such as POI and other exogenous factors such as chemotherapy. With advancements in array technologies, some key genes regulating follicle development, ovarian reserve, and POI have been reported (Table 1).

**Table 1 ijms-22-10819-t001:** List of genes related to premature ovarian inefficiency (POI) from previous studies.

Gene	Biological Category	Functions	Reference
*AMH*(19q13.3)	Anti-Müllerian hormone	Control of the formation of primary follicles by inhibiting excessive follicular recruitment by FSH	[21]
*AMHR2*(12q13)	AMH receptor	AMH signal transduction	[22]
*BMPR2*	BMP receptor	Signal transduction between oocytes and somatic cells	[23]
*BMP15*(Xq11.2)	Growth factor	Growth and differentiation of granulosa cells (GCs)	[24]
*ESR1*	Estrogen receptor	Regulation of follicle growth and maturation and oocyte release	[25]
*FIGLA*(2q13.3)	bHLH transcription factor	Regulation of multiple oocyte-specific genes, including genes involved in folliculogenesis and those that encode the zona pellucida	[26]
*FMR1*(Xq27)	Highly polymorphic CGG repeat in the 5′ untranslated region (UTR) of the exon 1	Transcriptional regulation	[27]
*FSHR*(2q21-p16)	Receptor	Follicular development and ovarian steroidogenesis	[28]
*FOXL2*(3q23)	Transcription factor	Differentiation and growth of granulosa cells	[29]
*FOXO3A*	Transcription factor	Regulating primordial follicle growth activation	[30]
*GDF9*(5q31.1)	Growth factor	Growth and differentiation of granulosa cells proliferation	[31]
*INHA* variants	Growth factor	Maturation of ovarian follicles by FSH inhibition	[32]
*KHDRBS1*	Signal transduction activator	Alter mRNA expression level and alternative splicing	[33]
*LHR*	Lutropin-choriogonadotropic hormone receptor	Regulation of ovarian follicle maturation, steroidogenesis, and ovulation	[34]
*LHX8*	Transcription factor	Germ-cell-specific critical regulator of early oogenesis	[35]
*NOBOX*(7q35)	Transcription factor	Follicle development	[36]
*PGRMC1*(Xq22-q24)	Heme-binding protein	Regulation of apoptosis	[37]
*POLR3H*	RNA polymerase III subunit H	Regulation of cell cycle, cell growth, and differentiation	[38]
*SOHLH1**	Transcription factor	Early folliculogenesis	[39]
*WT1*(11q13)	Transcription factor	Granulosa cell differentiation and oocyte-granulosa cell interaction	[40]

Especially, POI-related gene families and pathways, such as the Fas ligand (*FasL*)-Fas pathway, nuclear factor-kappa B (*NF-κB*), inhibitory kappa B α (*IκBα*), interleukin-1 receptor-associated kinase (*IRAK1*), and tumor necrosis factor receptor-associated factor 6 (*TRAF6*), have been reported in humans [41,42,43]. 

Other studies have demonstrated the relationship between mRNA-miRNA interaction during ovarian follicle development and POI regulation [44,45]. Correlations between miRNAs and POI have been studied, and miR-23a, miR-27a, miR-22-3p, miR-146a, miR-196a, miR-290-295, miR-423, and miR-608 have been suggested to be associated with this condition [5]. Among these miRNAs, miR-23a and miR-27a are involved in granulosa cell proliferation and apoptosis by controlling *XIAP* [6,7] and *IGFBP2* [8]. The apoptosis of granulosa cells is critical in POI, therefore, understanding the regulatory roles of miRNAs in granulosa cells enhances our understanding of the pathogenesis of POI [46]. 

The Piwil (Piwi) protein family is a small-RNA-bound effector complex [47,48], and the mouse genome encoded three Piwi proteins, PIWIL1/MIWI, PIWIL2/MILI, and PIWIL4/MIWI2 [49]. Mammalian Piwil proteins associate with Piwi-interacting RNAs (piRNAs) and piRNA expression is largely restricted to the germline [50]. The deficiency of genes required for piRNA biogenesis leads to infertility in males [51]; however, females with this genotype are not infertile [52]. Whereas the role of regulatory miRNAs in POI has been studied, their regulatory function in this condition is not well known.

In this study, we aimed to identify the miRNAs that regulate the ovarian age-specific genes, which might contribute to the regulation of POI. We also analyzed the correlation between Piwil and specific miRNAs.

## 2. Results

### 2.1. Identification of Transcripts in Different Reproductive Stages of Mouse Ovaries

We identified a total of 5058 DEGs (Figure 1 and Appendix A) including 2658 upregulated genes and 2400 downregulated genes in the reproductively aged mouse ovary based on comparisons among stages (Table 2). Correlations of total gene count (Figure 2A) from each stage and filtered count numbers were determined by respective FPKM and TPM analyses and displayed by heatmaps (Appendix A). The count data displayed by the PCA plot indicate the distance of expressed transcripts (Figure 2B). The distribution of transcripts is displayed as a cluster in Appendix A. The DEGs among the reproductively aged mouse ovaries were compared and different comparisons were divided into six groups as follows: (1) 6 weeks (W) vs. 2 W, (2) 15 W vs. 2 W, (3) 20 W vs. 2 W, (4) 15 W vs. 6 W, (5) 20 W vs. 6 W, (6) 20 W vs. 15 W. The DEGs are displayed as a volcano plot based on logFC values (Figure 3A–F). 

### 2.2. Gene Ontology (GO) Enrichment of DEGs

The overall biological processes associated with DEGs were analyzed, and the *p* value cutoff was set at <0.05. Gene ontology biological process, gene ontology molecular function, and gene ontology cellular component (GOCC) analyses were performed.

In the 6 W vs. 2 W group, the DEGs were mainly enriched in the steroid biosynthetic process, regulation of cholesterol metabolic process, organonitrogen compound biosynthetic process, and cholesterol biosynthetic process (Appendix A). In the 15 W vs. 2 W set, DEG sets comprising the regulation of cell migration, negative regulation of the cellular process, and extracellular matrix organization were enriched. Upregulated DEGs were mainly enriched in the sterol biosynthetic process, regulation of cholesterol metabolic process, and cholesterol biosynthetic process. Downregulated DEGs were enriched in the cellular protein metabolic process (Appendix A). Analysis revealed that lysosome, Golgi subcompartments, and Golgi membranes were enriched among cellular components (Appendix A). 

In the 20 W vs. 2 W set, DEGs were highly enriched in cytokine-medicated signaling pathways, and downregulated DEGs were mostly enriched in cellular protein metabolic process (Appendix A), neuropilin binding (Appendix A), and the endoplasmic reticulum lumen (Appendix A). In the 15 W and 6 W sets, DEGs were enriched in the cellular response to cytokine stimulus (Appendix A) and cytosolic part (Appendix A). In the 20 W vs. 6 W set, DEGs were enriched in the regulation of transcription from RNA polymerase II promoter, regulation of cell proliferation, and positive regulation of transcription (Appendix A). Furthermore, they were enriched in cytokine activity (Appendix A) and platelet alpha granules (Appendix A). For 20 W vs. 15 W, DEGs were enriched in the regulation of transcription from RNA polymerase II promoter and positive regulation of transcription (Appendix A). They were also enriched in RNA polymerase II regulatory region sequence-specific DNA binding (Appendix A). These data indicated that the transcriptomes in the ovary are altered and stage-specific DEGs are increased in the reproductively aged 20 W ovary.

### 2.3. KEGG Analysis of DEGs

The upregulated DEGs in each group were enriched in cytokine-cytokine receptors for 6 W vs. 2 W (Appendix A) and cell adhesion molecules for 15 W vs. 2 W (Appendix A). In the 20 W vs. 2 W set, DEGs were enriched in cytokine-cytokine receptors (Appendix A C). Hormones, steroids, ovarian signaling, and cortisol synthesis were upregulated in the 15 W vs. 6 W group (Appendix A). DEGs of 20 W vs. 6 W were enriched in TNF signaling, IL-17 signaling, MAPK signaling, and cytokine-cytokine receptor (Appendix A). TNF signaling and IL-17 signaling were highly enriched for DEGs that were upregulated (Appendix A).

### 2.4. Protein-Protein Interaction (PPI) Network Construction and Clusters Analyses

The PPI networks were built using STRING based on DEGs logFC values and are shown in Table 3 and Appendix A. The PPI networks of upregulated and downregulated DEGs were generated, and their biological processes, molecular functions, and KEGG pathways were analyzed. In the 6 W vs. 2 W set, *Klk1*, which has a role in the positive regulation of steroid hormone biosynthetic process, and the well-known *Figla* were downregulated (Table 3A and Appendix A). Steroid biosynthesis-related *Hsd17b7* and cholesterol metabolism-related *Star* were upregulated for 15 W vs. 2 W (Table 3B and Appendix A). The RNA polymerase regulator, *Atf3* was downregulated (Table 3B and Appendix A), and this phenomenon was also observed in the 15 W vs. 6 W set. Steroid hormone biosynthesis-related *Akr1c18* and *Hsd17b7* were also enriched in 20 W vs. 2 W (Table 3C and Appendix A). *Eif3j1*, which has translation initiation factor activity was upregulated for 15 W vs. 6 W (Table 3D and Appendix A). In the 20 W vs. 6 W set, *Oog1* was downregulated, and it is involved female gamete generation (Table 3E and Appendix A). For 20 W vs. 15 W, the IL-17 signaling pathway-related *Cxcl1* and *Atf3*, and the regulation of transcription from RNA polymerase II promoter were upregulated in response to endoplasmic reticulum stress (Table 3F and Appendix A).

### 2.5. Identification of Key Target Gene and Their Validation

Among the DEGs identified in each reproductive stage, we selected 90 highly expressed genes after filtering and selected 13, *Bmp15*, *Ccnd2*, *Figla*, *Foxo1*, *Foxo3*, *Fshr*, *Gdf9*, *Igf1*, *Inha*, *Nobox*, *Smad3*, *Taf4b*, and *Pou5f1*, as key target genes based on their correlation with the ovaries (Table 4). Their expression levels and ranked correlations were calculated and filtered using the FPKM and TPM values (*p* < 0.05). 

The expression levels of *Ccnd2* and *Igf1* were similar throughout the reproductively-aged mouse ovaries. The levels of *Figla*, *GDF9*, *Nobox*, and *Pou5f1* were decreased as reproductive aging progressed. *Foxo1*, *Inha*, and *Taf4b* demonstrated a more than two-fold increase in expression. Finally, in reproductively aged individuals, *Fshr* increased at 20 W. 

### 2.6. Correlation with miRNAs

Interacting miRNAs with the key target genes were identified based on the ranked correlation (*p* < 0.05) and are listed in Table 5. Their interactions with miRNAs were confirmed using databases (Appendix A). Continuously expressed genes, such as *Ccnd2* and *Igf1*, are target genes of various miRNAs (Table 5). Genes regulating oocyte and follicle development, such as *Figla*, *Foxo3*, and *Fshr*, were determined to interact with only one miRNA, mmu-miR-669d-2-3p, mmu-miR-153-3p, and mmu-miR-670-3p, respectively. 

In total, 27 genes were increased based on the regulatory effects of miRNAs, and 26 genes were decreased between the 2 W and 20 W sets (*p* < 0.05, Appendix A). During the reproductive period, some genes were downregulated following transient up-regulation. The genes expressed at high levels at 20 W and decreased at 2 W (Appendix A, left panel), as well as the genes expressed at high levels at 2 W and decreased at 20 W, set are listed in Appendix A, right panel. Genes upregulated following downregulation in the 20 W set (Appendix A, left panel) and 2 W set (Appendix A, right panel) are also listed with their interacting miRNAs.

Moreover, the up- and downregulated DEGs between 2 W and 20 W, and their correlations with miRNAs occurring in a reproductive-age-specific manner in the ovary were further analyzed using the database. mmu-miR-9-5p, mmu-miR-26a-5p, mmu-miR-29b-3p, mmu-miR-26a-5p, mmu-miR-706 were the miRNAs predominantly correlated with downregulated DEGs (Appendix A).

### 2.7. Interaction with Piwil Gene Family

The *Piwil* gene family is essential for developmental regulation. Therefore, we additionally analyzed the correlation between DEGs from reproductively aged mouse ovaries and this family (Appendix A). Target genes and each *Piwil* gene were analyzed and are listed in Appendix A. Target genes of *Piwil* included well-known early oocyte development and follicle assembly genes (Appendix A). *Piwil4* was determined to interact with mmu-miR-210-5p and mmu-miR-3470b, whereas *Piwil1* and *Piwil2* were found to interact with more miRNAs (Table 6).

## 3. Discussion

In this study, we demonstrated the stage-specific upregulation and downregulation of DEGs and their correlation with miRNAs in reproductively aged mouse ovaries. We identified 24,958 transcripts from each reproductive stage of ovaries, and 5058 genes were identified. In total, 2658 genes were upregulated and 2400 genes were downregulated. Furthermore, 53 upregulated and downregulated miRNAs were identified. GO analyses demonstrated that upregulated DEGs were enriched in different cellular processes in a stage-specific manner. DEGs from the early phase set of 2 W, ovaries were mainly enriched in the steroid biosynthetic process, regulation of cholesterol metabolic process, organonitrogen compound biosynthetic process, and cholesterol biosynthetic process. In ovaries from middle-phase sets, 6 W and 15 W, DEG sets associated with the regulation of cell migration, negative regulation of the cellular process, and extracellular matrix organization were enriched. KEGG analyses suggested that the upregulated genes were mostly enriched in cell adhesion molecules.

We also focused on the regulatory roles of miRNAs in reproductively aged mouse ovaries. The roles of miRNAs in folliculogenesis [4] and oocyte development have emerged [3]. We focused on the altered gene expression in a specific manner, and interactive regulatory miRNAs of target genes were identified based on a score > 90 (*p* <0.05). mmu-miR-136-5p, mmu-miR-335-5p, mmu-miR-665-3p, and mmu-miR-18a-5p targeted several upregulated genes in 20 W mouse ovaries (Appendix A). miRNAs targeted genes were involved in spermatogenesis, follicular development process. *ESR1*, *SMAD2*, and *Spata1* are the target of mmu-miR-18a-5p. *ESR1* has been involved in the genetic variation of female infertility, and *Sparta* is an acrosomal protein in sperm and related to genetic mutation in sperm. Its role on female infertility is still unknown. mmu-miR-136-5p targets *Prdm16* and *GDF6*, genes involved in follicular development. mmu-miR-335-5p targets *Calu*, regulator of chromosome condensation. The major target of mmu-miR-665-3p is *Pou2f3*, which belongs to same family of *Oct4*, master regulator of pluripotency.

In the down-regulated gene set, mmu-miR-9-5p, mmu-miR-26a-5p, mmu-miR-706, and mmu-miR-29b-3p correlated with several genes (Appendix A). mmu-miR-9-5p targets several fertility and germ cell developmental genes, such as *Frip2, Prdm6, Spam1, Prdm1, Sirt1* and *Map3k2*. mmu-miR-26a-5p mainly targets cell growth regulation by estrogen and voltage-dependent channel genes, *Kcnq4* and *Cacna1c*. mmu-miR-706 targets mesodermal genes and mmu-miR-29b-3p targets Bcl2 modifying factor (*Bmf*) and vascular endothelial growth factor A (*Vegfa*). In this finding, we hypothesize that sperm activity function regulating genes is a major target of up and downregulated miRNAs. Therefore, we conclude that several male fertility genes might be the involved onset of diminishing ovarian reserve.

Among the highly expressed DEGs, well-known ovary-associated genes were identified. The master regulator of oocyte development, *Figla* was downregulated in later-stage ovaries. *Figla* (factor in the germline alpha), encodes a germ cell-specific basic helix-loop-helix transcription factor first identified as an activator of oocyte genes [53,54]. *Figla* regulates primordial follicle formation in the fetal ovary [55] and antagonizes spermatogenesis during embryo development [56]. This gene is related to POI, and its expression was decreased according to reproductive age. Its interacting miRNA was mmu-miR-669d-2-3p, and the main function of this is to modify sperm-related gene expression [57]. This result supports the known function of *Figla* and supports its decreased expression throughout the reproductive span. 

*Foxo3* and *Fshr* are also mainly regulated by miRNAs. *Foxo3* is involved in the apoptosis of granulosa cells [58] and involved in the early development of follicles on its own [59] or with other genes [60]. *Foxo3* interacts with mmu-miR-153-3p, which is known to suppress tumor growth in ovarian carcinoma [61]. *Fshr* (follicle-stimulating hormone receptor) is one of the most important receptors in the ovary and other female reproductive organs. It is associated with the granulosa cells of the follicle and is located in the granulosa cells [62]. Interactive mmu-miR-670-3p is expressed in the newborn ovary [63,64], however, its roles are still not well known. 

*Klk1* has been involved in positive regulation of the steroid hormone biosynthetic process, is related to the renin-angiotensin system, and was enriched in the early and middle phase sets. The renin-angiotensin system, with angiotensin, is known for its relation to ovarian follicle development [65] and acquisition of dominancy [66]. Components are expressed in an estrogen-dependent manner in the uterus and are involved in cell proliferation [67]. Furthermore, they also involved the relationship with umbilical veins [68]. 

The *Piwil* family has been identified as comprising regulatory proteins responsible for stem cell and germ cell differentiation [69], and it is co-expressed with early developmental genes [70]. To date, the *Piwil* family is known to be more involved in male fertility than female fertility [50]. POI does not result in infertility from the beginning; therefore, we hypothesized that fine regulatory components might be involved in the onset of POI. Furthermore, they would be responsive to E2, which is another important molecule in the ovary [71]. The main targets of this family are enriched in genes such as *FOXL2*, *NANOG*, *POU5F1*, *LIN28A*, *SOX2*, *PRDM1*, *NANOS3*, *DAZL*, and *DDX4* (Table 6). These results indicate that Piwil might contribute to the regulation of certain DEGs in a stage-specific manner. 

Among this family, *Piwil4* interacted predominantly with two miRNAs, mmu-miR-210-5p, and mmu-miR-3470b. Further, mmu-miR-26a-5p, miRNA that was determined to interact with *Piwil2*, mostly interacted with downregulated genes, such as *APCDD1*, *CTDSP2*, *EZH2*, *MGA4A*, *PATZ1*, and *RHOQ*, based on the 2 W vs. 20 W comparison (Appendix A B). These results indicate that *Piwil2* and mmu-miR-26a-5p might be a candidate that regulates POI. Piwi-interacting RNAs (piRNAs) are involved in infertility in males, but not in females. We hypothesized that *Piwil2* does not lead to terminal infertility; however, it affects the ovarian reserve via downregulated genes. *APCDD1* is involved in adipogenic differentiation [72], *CTDSP2* is a target gene of *FOXO* and regulates cell cycle progression [72], *EZH2* is an important regulator in the female reproductive tract [73], *MGA4A* is involved in embryo lethality and female infertility [74], and *RHOQ* regulates mitochondrial function [75]. All genes regulated by miR-26a-5p were related to female infertility and cell cycle progression. Therefore, their correlation with POI needs to be further studied. These findings also correlated with the target genes regulated by mmu-miR-26a-5p were male fertility-related genes. 

In conclusion, we identified DEGs and interactive miRNAs in reproductively aged mouse ovaries. Stage-specific upregulation and downregulation of DEGs indicated the regulatory roles of miRNAs in the ovary and their correlation with ovarian reserve. We used three reproductively aged stages, which correspond to the reproductive span of a human. However, differences between species limit the interpretation of experimental results. POI is an endocrinological symptom only in women; therefore, identified miRNAs might have distinct roles in humans. Further studies are needed to elucidate the functions of these miRNAs as regulators of the ovarian reserve. We will further study the roles of these miRNAs during in vitro follicular development based on non-human primate models, which are physiologically closer to humans [76,77,78].

## 4. Materials and Methods

### 4.1. Ethics

All the experiments were conducted under the control of AAALAC guidelines, and the animal experimental plan was reviewed and approved by IACUC of Seoul National University Hospital (No.18-0029-C1A1).

### 4.2. Preparation of Ovaries and Extraction of Total RNA

The experimental scheme of this study is presented in Figure 1. Ovaries of 2, 8, 15, and 20 weeks old C57BL/6 strain female mice were collected immediately after cervical dislocation. The lipid pad near the ovaries was removed and the collected ovaries were washed with HBSS (Invitrogen, Waltham, MA, USA). And then, ovaries were minced into pieces by a surgical blade (Feather Safety Razor, Osaka, Japan). The pieces were digested with Collagenase Type IV (1 mg/mL, Invitrogen) at 37 °C for 45 min and centrifugated. Pelleted samples washed with PBS (Invitrogen) and total RNAs were extracted using Trizol kit (Invitrogen) according to manufacturer’s instruction. Extracted RNAs were quantified and qualified for further Total Omics Transcriptome analyses.

### 4.3. Total Transcriptome (RNA-Seq) Analyses

#### 4.3.1. mRNA Sequencing

The libraries were prepared for 150 bp paired-end sequencing using TruSeq Stranded mRNA Sample Preparation Kit (Illumina, CA, USA). Namely, mRNA molecules were purified and fragmented from 1μg of total RNA using oligo (dT) magnetic beads. The fragmented mRNAs were synthesized as single-stranded cDNAs through random hexamer priming. By applying this as a template for second-strand synthesis, double- stranded cDNA was prepared. After the sequential process of end repair, A-tailing, and adapter ligation, cDNA libraries were amplified with PCR (Polymerase Chain Reaction). The quality of these cDNA libraries was evaluated with the Agilent 2100 BioAnalyzer (Agilent, CA, USA). They were quantified with the KAPA library quantification kit (Kapa Biosystems, MA, USA) according to the manufacturer’s library quantification protocol. Following cluster amplification of denatured templates, sequencing was progressed as paired-end (2 × 150 bp) using Illumina NovaSeq 6000 sequencer (Illumina, CA, USA).

#### 4.3.2. microRNA Sequencing

The libraries were prepared for 50 bp single-end sequencing using the NEXTflex Small RNA-Seq Kit (Bioo Scientific Corp). Namely, small RNA molecules were isolated from 1 μg of total RNA via the adapter ligation. The isolated small RNAs were synthesized as single-stranded cDNAs through the RT (Reverse transcription) priming. By applying this as a template for second-strand synthesis, double-stranded cDNA was prepared through PCR (Polymerase Chain Reaction). And, the fragments around 150 bp were extracted for sequencing through size selection by gel electrophoresis. The quality of these cDNA libraries was evaluated with the Agilent 2100 BioAnalyzer (Agilent, CA, USA) followed by quantification with the KAPA library quantification kit (Kapa Biosystems, MA, USA) according to the manufacturer’s protocol. Following cluster amplification of denatured templates, sequencing was progressed as single-end (50 bp) using Illumina sequencing platform (Illumina, CA, USA).

### 4.4. Read Quality of the Experiment 

The identification was performed by the Sanger method, using Illumina 1.9. Total sequences were 32219029 before fast qc and 31058105 after fast qc. Sequence length ranged from 36 to 101 (Appendix A). The percentage of the GC was 49 and the GC content has not emerged from theoretical distribution (Appendix A). No rRNA contamination was observed. Sequences filtered as the poor quality was none (Appendix A) and the duplicated sequence was 36.32% (Appendix A).

### 4.5. Data Processing

We evaluated the data quality by sample clustering based on Pearson’s correlation matrices between different samples. And a heatmap was drawn corresponding to the different expressions of probes.

### 4.6. Differentially Expressed Genes (DEGs)

We employed the “limma” R language package to screen the DEGs between uterine leiomyoma and normal myometrium. The adjusted *p*-value < 0.05 and |log2fold change (FC)| > 1 were considered statistically significant.

### 4.7. KEGG and GO Enrichment Analyses of DEGs

Gene Ontology (GO, http://geneontology.org/, accessed on 1 February 2021) provides an ontology of defined terms to represent gene functions (molecular function, cellular component and biological process). Besides, Kyoto Encyclopedia of Genes and Genomes (KEGG, http://www.genome.jp/kegg/ accessed on 1 March 2021) is a database resource for understanding high-level functions and utilities of the biological system. A package in R language called “clusterprofiler” was used to determine the biological significance of DEGs. The package is capable of providing GO and KEGG enrichment analyses and visualization for users to obtain more valuable biological information [36]. *p* value < 0.05 was considered a significant enrichment.

### 4.8. Gene Set Enrichment Analysis (GSEA)

To explore the potential functions of selected key genes and microRNAs in mouse ovaries, and samples of datasets divided into different groups following the expression levels of the key genes, respectively. GSEA was utilized to explore whether priority determined biological processes datasets were enriched in these groups derived from DEGs between the two groups [30]. The criteria were set as *p*-value < 0.05 and FDR < 0.25.

### 4.9. miRNA Profiling

Profiling of expressed miRNAs and their target genes were annotated and analyzed by multiple databases: miRDB (http://mirdb.org, accessed on 1 January 2021), miRTarBase (https://bio.tools›mirtarbase, accessed on 1 December 2020), TarBase (https://carolina.imis.athena-innovation.gr, accessed on 10 December 2020), and TargetScan (http://www.targetscan.org, accessed on 20 December 2020).

### 4.10. Protein Interaction

Prediction of interaction with other functional genes was predicted by STRING database (https://string-db.org, accessed on 1 April 2021).

## Figures and Tables

**Figure 1 ijms-22-10819-f001:**
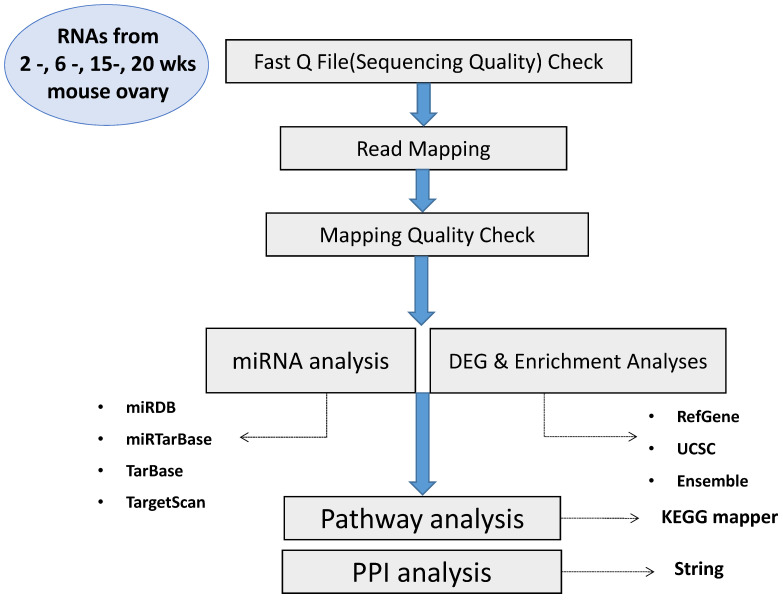
Experimental scheme of this study.

**Figure 2 ijms-22-10819-f002:**
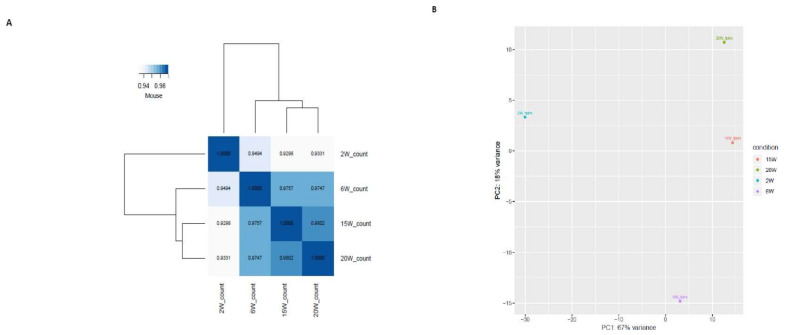
Correlation among analyzed genes in the reproductively aged mouse ovary. (**A**) Count data and correlation among each ovary stage, indicated by based on a heatmap, (**B**) Count data indicated by a PCA plot.

**Figure 3 ijms-22-10819-f003:**
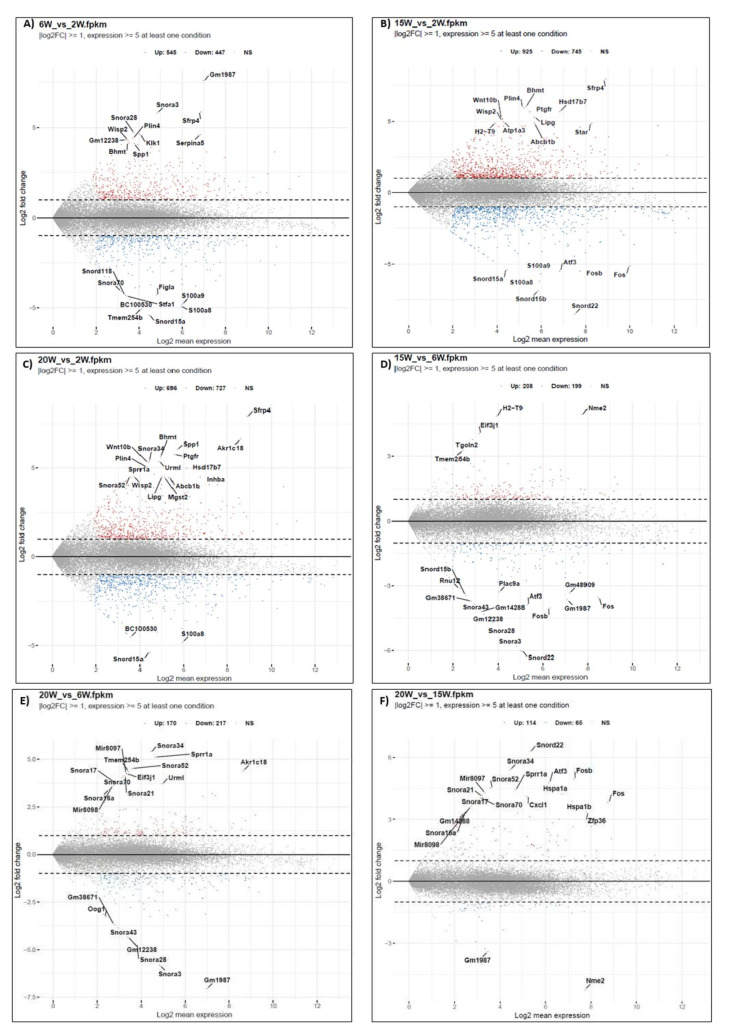
Up- and downregulated transcripts in the reproductively aged mouse ovary. Differently expressed transcripts among the differently aged mouse ovaries, based on the log2 fold changes and 5 fold up-and downregulation, filtered based on FPKM values. (**A**) 6 weeks (W) vs. 2 W, (**B**) 15 W vs. 2 W, (**C**) 20 W vs. 2 W, (**D**) 15 W vs. 6 W, (**E**) 20 W vs. 6 W, (**F**) 20 W vs. 15 W.

**Table 2 ijms-22-10819-t002:** Numbers of differentially expressed genes (DEGs) based on comparisons between the reproductive stages of the mouse ovary. (**A**) Total expression and upregulated and downregulated DEG numbers based on comparison for each aged mouse ovaries. (**B**) Quantitative graph of up- and downregulated DEG numbers.

A	6 W vs. 2 W	15 W vs. 2 W	20 W vs. 2 W	15 W vs. 6 W	20 W vs. 6 W	20 W vs. 15 W
Up-regulated	545	925	696	208	170	114
Down-regulated	447	745	727	199	217	65
Total expression	992	1670	1423	407	387	179
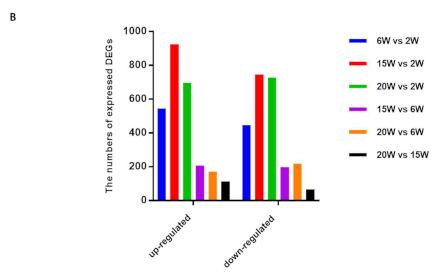						

**Table 3 ijms-22-10819-t003:** Predicted protein interaction for up- and downregulated transcripts in the reproductively aged mouse ovary. Table 2. Fold changes among the groups were selected and their predictive interaction was analyzed using the STRING database. Redundant genes were indicated when they first appeared in the table. (**A**) 6 weeks (W) vs. 2 W, (**B**) 15 W vs. 2 W, (**C**) 20 W vs. 2 W, (**D**) 15 W vs. 6 W, (**E**) 20 W vs. 6 W, (**F**) 20 W vs. 15 W.

**(A) 6 W vs. 2 W**		
**Up-Regulation**		
**Gene**	**Category**	**Biological Functions**
** *Bhmt* **	Biological process	Homocysteine catabolic process
	Molecular Function	Oxidoreductase activity, acting on the CH-NH group of donors, flavin as acceptor
	KEGG pathway	Cystein and methionine metabolism
** *Spp1* **	Biological process	Calcium-independent cell-matrix adhesion
	Molecular Function	Extracellular matrix binding
	KEGG pathway	ECM-receptor interaction
** *Klk1* **	Biological process	Positive regulation of steroid hormone biosynthetic process
	Molecular Function	Bradykinin receptor activity
	KEGG pathway	Renin-angiotensin system
** *Serpina5* **	Biological process	Blood coagulation, fibrin clot formation
	Molecular Function	Serine-type endopeptidase activity
	KEGG pathway	Complement and coagulation cascades
** *Sfrp4* **	Biological process	Positive regulation of dermatome development
	Molecular Function	Chemoattractant activity involved in axon guidance
	KEGG pathway	Basal cell carcinoma
** *Plin4* **	Biological process	Negative regulation of sequestering of triglyceride
	Molecular Function	Tryglyceride lipase activity
	KEGG pathway	Regulation of lipolysis in adipocytes
** *Wisp2* **	Biological process	Regulation of dermatome development
	Molecular Function	Fibronectin binding
	KEGG pathway	Bladder cancer
** *GM1987* ** ** *(Ccl21a)* **	Biological process	Lymphocyte chemotaxis across high endothelial venule
	Molecular Function	CXCR chemokine receptor binding
	KEGG pathway	Chemokine signaling pathway
**Down-Regulation**		
**Gene**	**Category**	**Biological Functions**
** *BC100530, Stfa1* **	Biological process	Negative regulation of endopeptidase activity
	Molecular Function	Cysteine-type endopeptidase inhibitor activity
** *Figla* **	Biological process	Negative regulation of fertilization
	Molecular Function	Acrosin binding
	KEGG pathway	Ovarian steroidogenesis
** *S100a9, S100a8* **	Biological process	Neutrophil aggregation
	Molecular Function	Toll-like receptor 4 binding
	KEGG pathway	IL-17 signaling pathway
**(B) 15 W vs. 2 W**		
**Up-Regulation**		
**Gene**	**Category**	**Biological Functions**
** *Wnt10b* **	Biological process	Wnt signaling pathway involved in midbrain dopaminergic neuron differentiation
	Molecular Function	Frizzled binding
	KEGG pathway	Basal cell carcinoma
** *H2-T9 (Gm7030)* **	Biological process	Antigen processing and presentation of exogenous protein antigen via MHC class
	Molecular Function	MHC class I protein binding
	KEGG pathway	Antigen processing and presentation
** *Atp1a3* **	Biological process	Positive regulation of sodium ion export across plasma membrane
	Molecular Function	Sodium:potassium-exchanging atpase activity
	KEGG pathway	Proximal tubule bicarbonate reclamation
** *Ptgfr* **	Biological process	Phospholipase c-activating angiotensin-activated signaling pathway
	Molecular Function	Angiotensin type I receptor activity
	KEGG pathway	Renin-angiotensin system
** *Lipg* **	Biological process	Triglyceride catabolic process
	Molecular Function	Lipoprotein lipase activity
	KEGG pathway	Glycerolipid metabolism
** *Abcb1b* **	Biological process	Drug export
	Molecular Function	Xenobiotic transmembrane transporting atpase activity
	KEGG pathway	Bile secretion
** *Hsd17b7* **	Biological process	Estrogen biosynthetic process
	Molecular Function	Estradiol 17-beta-dehydrogenase activity
	KEGG pathway	Steroid biosynthesis
** *Star* **	Biological process	Vesicle tethering to endoplasmic reticulum
	Molecular Function	Porin activity
	KEGG pathway	Cholesterol metabolism
** *Plin4, Bhmt, Sfrp4* **		Functions are indicated in previous page
**Down-Regulation**		
**Gene**	**Category**	**Biological Functions**
** *Atf3* **	Biological process	Positive regulation of transcription from RNA polymerase
	Molecular Function	Camp response element binding protein binding
	KEGG pathway	Cocaine addiction
** *Fos, Fosb* **	Biological process	Response to gravity
	Molecular Function	Nad-dependent histone deacetylase activity
	KEGG pathway	Amphetamine addiction
** *S100a9, S100a8* **		Functions are indicated in previous page
**(C) 20 W vs. 2 W**		
**Up-Regulation**		
**Gene**	**Category**	**Biological Functions**
** *Sprr1a* **	Biological process	Peptide cross-linking
** *Wisp2* **	Biological process	Regulation of dermatome development
	Molecular Function	Fibronectin binding
	KEGG pathway	Bladder cancer
** *Inhba* **	Biological process	Regulation of follicle-stimulating hormone secretion
	Molecular Function	Inhibin binding
	KEGG pathway	TGF-beta signaling pathway
** *Mgst2* **	Biological process	Xenobiotic catabolic process
	Molecular Function	Glutathione disulfide oxidoreductase activity
	KEGG pathway	Metabolism of xenobiotics by cytochrome P450
** *Akr1c18* **	Biological process	Polyprenol catabolic process
	Molecular Function	Enone reductase activity
	KEGG pathway	Steroid hormone biosynthesis
** *Sfrp4* **	Biological process	Positive regulation of dermatome development
	Molecular Function	Chemoattractant activity involved in axon guidance
	KEGG pathway	Basal cell carcinoma
** *Wnt10b, Plin4, Lipg, Bhmt,* ** ** *Abcb1b, Spp1, Ptgfr, Hsd17b7* **		Functions are indicated in previous page
**Down-Regulation**		
**Gene**	**Category**	**Biological Functions**
** *BC100530* **	Biological process	Negative regulation of endopeptidase activity
	Molecular Function	Cysteine-type endopeptidase inhibitor activity
** *S100a8* **		Functions are indicated in previous page
**(D) 15 W vs. 6 W**		
**Up-Regulation**		
**Gene**	**Category**	**Biological Functions**
** *Tgoln1* ** ** *(Tgoln2)* **	Biological process	Golgi ribbon formation
	Molecular Function	Mannose binding
	KEGG pathway	SNARE interactions in vesicular transport
** *Eif3j1* **	Biological process	Viral translational termination-reinitiation
	Molecular Function	Translation initiation factor activity
	KEGG pathway	RNA transport
** *Nme2* **	Biological process	ITP metabolic process
	Molecular Function	CTP synthase activity
	KEGG pathway	Pyrimidine metabolism
** *Tmem254b* **		Unknown
** *H2-T9 (Gm7030)* **		Functions are indicated in previous page
**Down-Regulation**		
**Gene**	**Category**	**Biological Functions**
** *Plac9a* **		Unknown
** *Atf3, Fosb,* ** ** *GM1987 (Ccl21a), Fos* **		Functions are indicated in previous page
**(E) 20 W vs. 6 W**		
**Up-Regulation**		
**Gene**	**Category**	**Biological Functions**
** *Tmem254b* **		Unknown
** *Eif3j1, Sprr1a* **		Functions are indicated in previous page
**Down-Regulation**		
**Gene**	**Category**	**Biological Functions**
** *Ooog1* **	Biological process	Female gamete generation
** *GM1987 (Ccl21a)* **		Functions are indicated in previous page
**(F) 20 W vs. 15 W**		
**Up-Regulation**		
**Gene**	**Category**	**Biological Functions**
** *Cxcl1* **	Biological process	Interleukin-8-mediated signaling pathway
	Molecular Function	Interleukin-8 receptor activity
	KEGG pathway	IL-17 signaling pathway
** *Atf3* **	Biological process	Positive regulation of transcription from RNA polymerase II promoter in response to endoplasmic reticulum stress
	Molecular Function	Camp response element binding protein binding
	KEGG pathway	Cocaine addiction
** *Hspa1a, Hspa1b* **	Biological process	Telomerase holoenzyme complex assembly
	Molecular Function	CTP binding
	KEGG pathway	Prion diseases
** *Zfp36* **	Biological process	Epithelial cell proliferation involved in salivary gland morphogenesis
	Molecular Function	Mrna 3’-UTR au-rich region binding
	KEGG pathway	Antifolate resistance
** *Sprr1a, Fosb, GM1987 (Ccl21a), Nme2* **		Functions are indicated in previous page
**Down-Regulation**		
**Gene**	**Category**	**Biological Functions**
** *GM1987 (Ccl21a), Nme2* **		Functions are indicated in previous page

**Table 4 ijms-22-10819-t004:** Key regulatory genes expressed in the reproductively-aged mouse ovary, analyzed by FPKM and TPM.

Genes		*Bmp15*	*Ccnd2*	*Figla*	*Foxo1*	*Foxo3*	*Fshr*	*Gdf9*	*Igf1*	*Inha*	*Nobox*	*Smad3*	*Taf4b*	*Pou5f1*
**2 W**	** *FPKM* **	14.64	44.14	51.84	19.5	10.55	3.07	130.57	15.29	533.01	33.33	28.04	3.82	24.29
**6 W**		20.02	44.18	1.68	66.92	7.96	8.71	52.19	16.04	1396.52	7.42	16.71	4.5	5.17
**20 W**		9.55	47.87	1.8	82.89	10.89	21.87	33.55	16	1915.61	6.47	26.57	9.52	3.62
**2 W**	** *TPM* **	20.33	61.29	71.97	27.08	14.65	4.27	181.28	21.23	740.04	46.27	38.93	5.31	33.73
**6 W**		31.03	68.48	2.61	103.73	12.34	13.49	80.89	24.87	2164.77	11.51	25.91	6.97	8.02
**20 W**		16.46	82.45	3.09	142.78	18.75	37.68	57.79	27.55	3299.65	11.14	45.76	16.4	6.23

**Table 5 ijms-22-10819-t005:** Correlated miRNAs that interact with key target genes.

	Gene	Mature miRNAs
**FPKM**	** *Ccnd2* **	mmu-miR-1251-5p, mmu-miR-503-5p, mmu-miR-497a-5p, mmu-miR-195b, mmu-miR-322-5p, mmu-miR-15a-5p, mmu-miR-16-5p, mmu-miR-195a-5p, mmu-miR-195a-5p, mmu-miR-6955-3p, mmu-miR-29a-3p, mmu-miR-29b-3p, mmu-miR-29c-3p, mmu-miR-302c-3p, mmu-miR-302b-3p, mmu-miR-302d-3p, mmu-miR-302a-3p, mmu-miR-294-3p, mmu-miR-295-3p, mmu-miR-291a-3p, mmu-miR-106a-5p, mmu-miR-106b-5p, mmu-miR-93-5p, mmu-miR-20a-5p, mmu-miR-20b-5p, mmu-miR-17-5p, mmu-miR-19b-3p, mmu-miR-19a-3p, mmu-miR-503-5p, mmu-miR-195a-5p, mmu-miR-16-5p, mmu-miR-195b, mmu-miR-322-5p, mmu-miR-15a-5p, mmu-miR-497a-5p, mmu-miR-322-5p, mmu-miR-195a-5p, mmu-miR-195b, mmu-miR-15a-5p, mmu-miR-16-5p, mmu-miR-15b-5p, mmu-miR-503-5p, mmu-miR-182-5p, mmu-miR-9-3p, mmu-miR-124-3p.1, mmu-miR-1193-3p
	** *Figla* **	mmu-miR-669d-2-3p
	** *Foxo1* **	mmu-miR-370-3p, mmu-miR-183-5p, mmu-miR-183-5p.2, mmu-miR-96-5p, mmu-miR-144-3p, mmu-miR-27a-3p, mmu-miR-27b-3p, mmu-miR-411-5p, mmu-miR-145b, mmu-miR-145a-5p, mmu-miR-6715-5p, mmu-miR-135a-5p, mmu-miR-135b-5p
	** *Foxo3* **	mmu-miR-153-3p
	** *Fshr* **	mmu-miR-670-3p
	** *Igf1* **	mmu-miR-365-3p, mmu-miR-1a-3p, mmu-miR-206-3p, mmu-miR-483-3p.2, mmu-miR-18a-5p, mmu-miR-18b-5p, mmu-miR-452-5p, mmu-miR-29c-3p, mmu-miR-29b-3p, mmu-miR-29a-3p, mmu-miR-495-3p, mmu-miR-378c, mmu-miR-378a-3p, mmu-miR-1839-5p, mmu-let-7a-5p, mmu-let-7c-5p, mmu-miR-98-5p, mmu-let-7i-5p, mmu-let-7g-5p, mmu-let-7d-5p, mmu-let-7b-5p, mmu-let-7k, mmu-let-7f-5p, mmu-let-7e-5p, mmu-miR-192-5p, mmu-miR-215-5p, mmu-miR-29b-3p, mmu-miR-29a-3p, mmu-miR-29c-3p, mmu-miR-142a-5p, mmu-miR-340-5p, mmu-miR-489-3p, mmu-miR-425-5p, mmu-miR-186-5p
	** *Smad3* **	mmu-miR-129-2-3p, mmu-miR-129-1-3p, mmu-miR-145a-5p, mmu-miR-145b, mmu-miR-3154
	** *Taf4b* **	mmu-miR-148a-3p, mmu-miR-152-3p, mmu-miR-148b-3p
	** *Pou5f1* **	mmu-miR-218-5p, mmu-miR-1955-5p, mmu-miR-881-3p, mmu-miR-186-5p

**Table 6 ijms-22-10819-t006:** Correlation of *Piwil* gene family expression and frequently expressed target genes and microRNAs in the reproductively aged mouse ovary. Target genes of the *Piwil* gene family and interactive miRNAs of each *Piwil* gene were analyzed based on the FPKM correlation (*p* < 0.05, score 90+).

	**Gene**	**Target Genes and Mature miRNAs**
**FPKM**	** *Target genes* **	
	FOXL2, FOXO3, NANOG, POU5F1, LIN28A, SOX2, SOX3, KLF4, PRDM1, NANOS3, UTF1, CD38, DAZL, DDX4, SYCP3, RNF17, TDRD5, TDRD9, SOX17, GATA4, TEAD4, TFAP2C, LHX9, WT1	
	** *Interactive miRNAs* **	
	** *Piwil1* **	mmu-miR-1249-5p, mmu-miR-147-5p, mmu-miR-1892, mmu-miR-3084-3p, mmu-miR-330-5p, mmu-miR-6540-3p, mmu-miR-7116-3p, mmu-miR-7669-5p
	** *Piwil2* **	mmu-miR-103-2-5p, mmu-miR-1198-5p, mmu-miR-139-5p, mmu-miR-145a-5p, mmu-miR-149-5p, mmu-miR-1b-5p, mmu-miR-216b-5p, mmu-miR-223-5p, mmu-miR-23a-3p, mmu-miR-26a-5p, mmu-miR-26b-3p, mmu-miR-346-5p, mmu-miR-425-3p, mmu-miR-466d-3p, mmu-miR-466f-3p, mmu-miR-466g, mmu-miR-7065-3p, mmu-miR-760-3p, mmu-miR-7647-3p, mmu-miR-7661-5p, mmu-miR-935
	** *Piwil4* **	mmu-miR-210-5p, mmu-miR-3470b

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
