# Peer review of "Transcriptome Analyses Identify Potential Key microRNAs and Their Target Genes Contributing to Ovarian Reserve"

_ijms, 2021, doi:10.3390/ijms221910819_

Round 1
Reviewer 1 Report
This original research manuscript that addresses the expression of miRNA in premature ovarian insufficiency is a resubmitted manuscript which I have also reviewed in the previous evaluation.
The manuscript improved the information available on results and methods, but has put significantly less effort in the improvement of the theory supporting their research proposal and discussing their results.
Specific issues with the manuscript in its present form:
-
the abstract is too long with too little substantive matter: although the journal does not use the IMRD format for abstracts from the content matter try to summarize more the main methods and findings in the summary. It is obsolete to explain what miRNAs are in the summary.
-
the introduction needs an essential literature overview on the available miRNAs (in detail, as is currently provided in the discussion section 2nd paragraph). In the introduction the state-of-art needs to be reported in enough detail in order to allow the reader to understand, in light of the reported state of the art, what is the added value of this work. So if the work add to the body of knowledge.
-
the discussion needs to report in detail the results - a well developed introduction with a state of the art would allow the researchers to appropriately develop statements such as: Piwil2, mmu-miR-26a-5p demonstrated as most abundant interactive ones with down-regulated genes, APCDD1, CTDSP2, EZH2, MGA4A, PATZ1, RHOQ,
in comparison of 2W vs 20W set and would and evaluate well enough what Piwil2 and miR-26-5p would mean for the understanding of POI in the future.
I believe this work needs a major overhaul and comparison with the state of the art and a thoughtful discussion of the results in order to contribute to the current body of knowledge.
Reviewer 2 Report
Minor point:
“1. Introduction”, Page 2/19, Lines 62-65:
“miRNAs are small non-coding RNAs that contribute to the regulation of gene expression at transcription and post-transcription [12,13]. They are conserved in vertebrates [14] and rules numerous cellular functions including embryonic development [15] and other biological processes [13,16]”.
At that point, you can also report that differential expression of miRNAs has been studied in either endometriosis or ovarian cancer. Several differentially expressed miRNA in endometriosis compared to ovarian cancer have been found, mainly linked with epithelial–mesenchymal transition. Two common miRNAs overexpressed in both diseases were miR-325 and miR-492. While the expression of miR-325 was upregulated in both diseases, this was more prominent in ovarian cancer, suggesting that miR-325 could have a role in the transition from endometriosis to ovarian cancer.
Recommended reference: Samartzis EP, et al. Endometriosis-associated ovarian carcinomas: insights into pathogenesis, diagnostics, and therapeutic targets—a narrative review. Ann Transl Med 2020;8(24):1712.
Round 2
Reviewer 1 Report
This manuscript has improved in its quality from the first submission and I have only a few additional comments to the manuscript, which however warrant a major overhaul.
The manuscript is better served if the first two paragraphs of the discussion are moved to the introduction:
"Diminished ovarian reserve such as POI is a rising problem and leads to infertility of women under age 40. Throughout the previous studies, genes involved with POI have been known (Table 1), still critical causes or genes of POI have not been fully elucidated. Besides the regulatory genes, non-coding RNAs are considered as candidate one as regulators.
microRNAs are the well-known conserved regulator of gene expression [12], and their contribution to germ cell development control has been studied. Their contribution to ovarian follicles and oocyte development was also demonstrated [17,59,60]. The correlation of miRNAs to POI has been studied and several miRNAs have been suggested to be associated with POI [47]. Among these miRNAs, miR-23a and miR-27a are involved in granulosa cell proliferation and apoptosis by controlling XIAP [48,49] and IGFBP2 [50]."
DISCUSSION
- you should start a discussion with your findings, the paragraph In this study ... is an appropriate start
- Expain what the biological role in the context of fertility is more in detail for the miR which you discovered are important in POI - only so can you give context. Without this you do not add new knowledge. Just listing the findings again in the discussion section will not yield a better understanding of this topic - you need to explain it in line with the biological role that is already known for this in infertility more in detail
I do believe a linguistic check to better elevate the understanding of this work is also needed
Round 3
Reviewer 1 Report
My comments have been addressed. A minor linguistic check is still needed but overall the manuscript has highly improved in quality.
This manuscript is a resubmission of an earlier submission. The following is a list of the peer review reports and author responses from that submission.
Round 1
Reviewer 1 Report
The authors analyzed the mouse model ovarian tissue on differential gene expression in order to improve our understanding of ovarian reserve regulation and consequentially premature ovarian insufficiency. The manuscript needs major improvements prior to further review.
General remarks:
- Make sure, it is clear that this data is from a mouse model, to prevent that the reader is mislead
- Give more context to the clinical problem this addresses
Abstract:
- Shorten the abstract so that it contains only the vital information. Currently it reads as an introduction. Lines 11-15 should be shortened to only tell us why POI is a problem. The same remark is also valid from 18 – 23, where just state why miRNAs are important in POI.
- State in the begginings of the method description »In 24 this study, we aimed to identify key microRNAs and their target genes for the reproductive span of 25 the ovary. RNA sequencing was performed of 4 stages, 2 weeks (2W), 6 - (6W), 15- (15W), and 20- 26 (20W), of ovarian tissue samples and analyzed using public databases. The 5058 genes were identi- 27 fied as differentially expressed genes (DEGs) among the 2W, 6W, 15W, and 20W samples of mouse 28 ovaries. That this research has been done in an animal model. Also, please summarize and do not repeat 2 times for which timeframes you performed the analysis.
Introduction:
- The lines 49 – 52 make no sense – what is the topic of your research, if you are focusing on POI, define POI in the beginning and focus your introduction on POI
- In lines 72 – 73 you point out, that research focuses on major gene families: which are these gene families, how much connection has already been established regarding these gene families?
- The introduction is very scarce and would benefit of a figure or a list of currently known gene families associated with POI.
- What was the main research question of your study? Finish the introduction with the main research question please
Results:
- The lines 125 – 129 should be part of methodology
- The text following Table 1, figure 2 and figure 3 is confusing. Please try to improve language clarity. Also, figure 1 is missing or is not labeled
- In general the page 6/19 needs to be more clearly written on what exactly were the main findings – the reader has difficulties in understanding the main outcomes
- Table 2 needs rework: what do the columns means, is it function? Give column labels for clarity
- You give nowhere an explanation, what the difference between Key target genes and Target genes is? Why do you make this distincition, what is the point of it? If you want to focus only on key target genes then please do so and do not write additionally about the target genes (maybe add these in supplemental data)
- I would suggest that the miRNA are added in supplemental data (table 5)
- Consider a similar approach to table 6 – and write up the important data with a reference to the supplemental data or consider a more clear depiction of the data in table 6
- If you want to analyze the Piwil gene family in correlations with DEGs please be sure to introduce this earlier in the introduction and explain why this is important to compare within results.
Discussion:
- The authors started this paper with a very clinical question of factors impacting POI, but while identifying several differentially expressed genes, no further correlation to the current body of knowledge on POI and differential gene expression is done. Revise the discussion accordingly.
- Discuss the limitations this data has if we compare it to human ovarian tissue
- Clearly define what the further routes of research and their implications are based on your data
Reviewer 2 Report
In this article, the authors demonstrated the regulatory roles of miRNAs in ovarian reserve by targeting genes in a reproductive stage-specific manner. The manuscript is straightforward, well written, and concise and has clear results within the scope of a review article. Definitely deserves to be published and is a valuable contribution to the “International Journal of Molecular Sciences”.
Minor point:
“1. Introduction”, Page 2/19, Lines 62-65:
“miRNAs are small non-coding RNAs that contribute to the regulation of gene expression at transcription and post-transcription [12,13]. They are conserved in vertebrates [14] and rules numerous cellular functions including embryonic development [15] and other biological processes [13,16]”.
At that point, you can also report that differential expression of miRNAs has been studied in either endometriosis or ovarian cancer. Several differentially expressed miRNA in endometriosis compared to ovarian cancer have been found, mainly linked with epithelial–mesenchymal transition. Two common miRNAs overexpressed in both diseases were miR-325 and miR-492. While the expression of miR-325 was upregulated in both diseases, this was more prominent in ovarian cancer, suggesting that miR-325 could have a role in the transition from endometriosis to ovarian cancer.
Recommended reference: Samartzis EP, et al. Endometriosis-associated ovarian carcinomas: insights into pathogenesis, diagnostics, and therapeutic targets—a narrative review. Ann Transl Med 2020;8(24):1712.